# One-Year Outcome of Glycoprotein IIb/IIIa Inhibitor Therapy in Patients with Myocardial Infarction-Related Cardiogenic Shock

**DOI:** 10.3390/jcm10215059

**Published:** 2021-10-29

**Authors:** Krzysztof Myrda, Mariusz Gąsior, Dariusz Dudek, Bartłomiej Nawrotek, Jacek Niedziela, Wojciech Wojakowski, Marek Gierlotka, Marek Grygier, Janina Stępińska, Adam Witkowski, Maciej Lesiak, Jacek Legutko

**Affiliations:** 1Silesian Center for Heart Diseases, 3rd Department of Cardiology, 41-800 Zabrze, Poland; m.gasior@op.pl (M.G.); jacek.niedziela@gmail.com (J.N.); 23rd Department of Cardiology, Faculty of Medical Sciences in Zabrze, Medical University of Silesia, 40-055 Katowice, Poland; 3Institute of Cardiology, Jagiellonian University Medical College, 31-088 Kraków, Poland; mcdudek@cyfronet.pl (D.D.); jlegutko@kcri.org (J.L.); 4GVM Care & Research, Maria Cecilia Hospital, Cotignola, 48033 Ravenna, Italy; 5Clinical Department of Interventional Cardiology, John Paul II Hospital, 31-202 Kraków, Poland; nawrotek@gmail.com; 6Department of Cardiology and Structural Heart Diseases, 3rd Division of Cardiology, Medical University of Silesia, 40-055 Katowice, Poland; wojtek.wojakowski@gmail.com; 7Department of Cardiology, Institute of Medicine, University of Opole, 45-040 Opole, Poland; marek.gierlotka@gmail.com; 81st Department of Cardiology, Poznan University of Medical Sciences, 61-701 Poznań, Poland; marek.grygier@skpp.edu.pl (M.G.); maciej.lesiak@skpp.edu.pl (M.L.); 9Institute of Cardiology, 04-628 Warszawa, Poland; janina@stepinska.pl.pl; 10Department of Interventional Cardiology and Angiology, National Institute of Cardiology, 04-628 Warszawa, Poland; witkowski@hbz.pl

**Keywords:** acute coronary syndrome, cardiogenic shock, glycoprotein IIb/IIIa receptor inhibitors, myocardial infarction, percutaneous coronary intervention

## Abstract

Background: We aimed to evaluate the effect of intravenous glycoprotein IIb/IIIa receptor inhibitors (GPIs) on in-hospital survival and mortality during and at the 1-year follow-up in patients undergoing percutaneous coronary intervention (PCI) for myocardial infarction (MI) complicated by cardiogenic shock (CS), who were included in the Polish Registry of Acute Coronary Syndromes (PL-ACS). Methods: From 2003 to 2019, 466,566 MI patients were included in the PL-ACS registry. A total of 10,193 patients with CS received PCI on admission. Among them, GPIs were used in 3934 patients. Results: The patients treated with GPIs were younger, had lower systolic blood pressure on admission, required inotropes and intra-aortic balloon pump (IABP) support more frequently, and showed a lower efficacy of coronary angioplasty. In both groups, the same rates of in-hospital adverse events were observed. A lower mortality rate was reported in the group treated with GPIs 12 months after admission (54.9% vs. 57.9%, *p* = 0.002). Therapy with GPI was an independent factor reducing the risk of mortality in the 12-month follow-up. Conclusions: The addition of GPIs to the standard pharmacotherapy combined with PCI in patients with MI and CS on admission reduced the risk of death in the 12-month follow-up period without increasing in-hospital adverse event rates.

## 1. Introduction

According to the available knowledge and current guidelines, the management of patients with myocardial infarction (MI)-related cardiogenic shock (CS) should first include an early invasive strategy with restoration of infarct-related artery (IRA) patency. The implementation of this strategy significantly reduces the risk of in-hospital and long-term mortality [1,2,3]. The mortality rate still remains at an unacceptably high level despite the successful systemic reduction in time from diagnosis to restoration of flow in IRA, the preference for radial access, the usage of subsequent generations of drug-eluting stents (DES), the implementation of short-term left ventricular assist devices (LVAD), and therapeutic hypothermia [2,3,4,5,6,7,8].

Antiplatelet drugs are crucial in MI treatment [4,5]. Pharmacokinetic and pharmacodynamic disorders of orally administered drugs that occur during CS can translate into poorer clinical outcomes [7]. It can be caused by impaired absorption in the gastrointestinal tract, which is exacerbated by morphine use, or inefficient conversion of the prodrug to the active form in the liver in hypotonia. The increase in the doses of drugs to achieve therapeutic activity may in turn translate into a higher risk of adverse events [6]. In such cases, intravenous medications such as cangrelor or glycoprotein IIb/IIIa receptor inhibitors (GPIs) may be desirable adjuncts [2,3,9,10]. According to the current recommendations, the use of GPIs should be restricted to selected clinical situations. The knowledge of the platelet GPIs for the treatment of MI-related CS patients remains limited to a few non-randomized observational trials. In two of the largest observational studies, adding abciximab or eptifibatide to the therapy was associated with a lower risk of death in the short-term and during the 1-year follow-up [11,12].

Difficulties related to the treatment of patients with CS and a further need for the optimization of treatment due to high mortality rates of patients with CS in the course of MI prompted us to evaluate the therapeutic effect of GPIs in this group of patients. The reason for conducting this analysis was the availability of extensive data on patients, with MI collected as part of the Polish Registry of Acute Coronary Syndromes (PL-ACS). The main aim of the analysis was to search for the relationship between intravenous GPIs and in-hospital survival during the 1-year follow-up and possible in-hospital adverse events that could be due to the use of this group of drugs.

## 2. Materials and Methods

### 2.1. The PL-ACS Registry

The PL-ACS registry design was previously reported [13]. In brief, the PL-ACS registry is a nationwide multicenter prospective observational study of consecutively hospitalized patients with ACS. This registry was a joint initiative of the Silesian Center for Heart Diseases in Zabrze and the Polish Ministry of Health. Logistic support was granted from the National Health Fund, a nationwide public health insurance institution providing obligatory policies for all Polish citizens. The pilot phase of the registry started in October 2003. From June 2005, all Polish regions collected the data for the registry. Overall, 417 centers participated in the registry, including 59 centers (14%) with on-site catheterization facilities. A detailed protocol was prepared before starting the registry with inclusion and exclusion criteria, methods, logistics, and definitions for all fields in the data set. In May 2004, the definitions were adapted accordingly to be compatible with the Cardiology Audit and Registration Data Standards [14]. According to the protocol, all admitted patients with suspected ACS were screened to be eligible for the registry. However, the final enrollment proceeded after confirmation of ACS. The initial diagnosis was made by the attending physician based on clinical presentation, initial electrocardiographic pattern, and markers of myocardial necrosis. The patients were classified as having unstable angina, non–ST-segment elevation MI, or ST-segment elevation MI. If patients were hospitalized during the same ACS events in more than 1 hospital (transferred patients), all hospitals were required to complete the case report form. These hospitalizations were linked together during data management and were analyzed as 1 case of ACS. The data were collected by attending physicians and entered directly into the electronic case report form, or a printed case report form was used temporarily before converting the data into the electronic version. Internal data checks were implemented by the software. All-cause mortality data, including exact dates of death, were obtained from the National Health Fund and analyzed at the Silesian Center for Heart Diseases—the data management and analysis center. The registry was approved by the local ethics committee and met the conditions of the Declaration of Helsinki.

### 2.2. Patients and Definitions

The analysis was conducted in consecutive patients with STEMI and NSTEMI who underwent percutaneous coronary intervention (PCI) as the final method of reperfusion treatment with CS symptoms on admission. The patients were included in the PL-ACS registry between 1 October 2003 and 1 August 2019. The patients who developed CS in the course of mechanical complications of MI were excluded from further analysis. The diagnosis of STEMI or NSTEMI was based on the valid definitions of MI in the appropriate period of time. CS in PL-ACS was diagnosed based on a generally accepted definition as systolic blood pressure < 90 mmHg or the use of catecholamine therapy to maintain systolic pressure of at least 90 mmHg, clinical signs of pulmonary congestion, and signs of impaired organ perfusion with at least one of the following manifestations: altered mental status, cold and clammy skin and limbs, oliguria with a urine output < 30 mL per hour, or an arterial lactate level > 2.0 mmol per liter [1]. The invasive strategy was defined as the performance of coronary intervention during the index hospitalization. Flow in the epicardial artery was assessed using the TIMI scale. Decisions related to treatment modalities (i.e., the use of stents, intra-aortic balloon pump [IABP], intravenous GPIs, methods of angioplasty) were left to the discretion of the attending physicians. In-hospital and long-term complications were defined as follows: death—death from all causes (cardiac and noncardiac); reinfarction—an ischemic event that met the ESC and American College of Cardiology (ACC) criteria for infarction and was clearly clinically distinct from the index event at the time of admission [15]; stroke (hemorrhagic or ischemic)—acute neurologic deficit that lasted >24 h and affected the ability to perform daily activities or resulted in death. Major bleeding was defined as bleeding (I) associated with >5 g/dL (0.5 g/L) decrease in the hemoglobin level or >15% (absolute) decrease in the hematocrit level, (II) the event that caused hemodynamic compromise, or (III) the requirement for blood transfusion; resuscitated cardiac arrest was defined as sudden cardiac arrest due to ventricular fibrillation, ventricular tachycardia, electromechanical dissociation, or asystole. The vital status at 12 months was obtained from the official mortality records from the government databases and was available for all patients with the exact date of death.

The patients with MI complicated by CS undergoing PCI were divided into two groups depending on the use of GPIs. The outcome measures we analyzed included in-hospital major cardiac adverse events (death from all causes, reinfarction, stroke, major bleeding, cardiac arrest) and 12-month mortality. The study flow chart is given in Figure 1.

### 2.3. Statistical Analysis

Baseline demographic and clinical characteristics, angiographic findings, in-hospital adverse events, drugs at discharge, and mortality in the 12-month follow-up were compared depending on the use of GPIs. Continuous variables were summarized using arithmetic mean with standard deviation (SD) for normal distribution or median with interquartile range (IQR) for non-normal distribution. Normality of distribution was verified using the Shapiro–Wilk test. Continuous variables following normal distribution were compared using Student’s *t*-test, whereas variables other than normal were compared using the Mann–Whitney U test. Categorical variables were summarized using frequency tables. The chi-squared test with Yates’s modification was used for the comparison of categorical data, if applicable. Long-term survival was compared using the log-rank test and the Kaplan–Meier model was used to present cumulative survival probability. The multivariable analysis of factors affecting 12-month mortality was performed. Forward stepwise logistic regression with cross validation was used. The multivariable model for 12-month mortality included over 30 variables, i.e., age, sex, years of hospitalization, history of coronary artery disease (CAD), diabetes mellitus (DM), hypertension, hyperlipidemia, chronic heart failure (CHF), atrial fibrillation, previous stroke, presence of peripheral artery disease (PAD), chronic kidney disease (CKD), previous MI, previous PCI, previous CABG, sudden cardiac arrest before admission, type of MI (STEMI vs. NSTEMI), smoking status, systolic blood pressure (SBP) and heart rate (HR) on admission, number of significant stenoses on coronarography, significant left main (LM) disease, thrombolysis in MI (TIMI) flow after PCI, left ventricular ejection fraction (LVEF), in-hospital medication, including type of antiplatelet drug (ticlopidine, clopidogrel, prasugrel and ticagrelor), use of GPIs, IABP or inotropic support, and in-hospital adverse events such as stroke, cardiac arrest, subsequent MI, or bleeding requiring blood transfusion.

The assessment of the predictive power of the multivariable model (the relationship between the use of GPIs in patients with CS on admission and the annual mortality) was performed using the ROC curve model (Figure 2).

The results of the multivariable analysis were summarized as odds ratios (ORs) with 95% confidence intervals (CIs). The results were considered statistically significant for *p* < 0.05. Calculations were undertaken using STATISTICA PL version 13.3 (TIBCO, Palo Alto, Santa Clara, CA, USA).

## 3. Results

### 3.1. Baseline Characteristics

Finally, 10,193 MI patients with CS on admission were enrolled in the analysis. In view of their clinical status, all patients underwent coronary angiography and percutaneous revascularization mostly in the course of ST-segment elevation MI (79.1%). GPIs were used in 38.6% of patients from the study group. Of GPIs, eptifibatide was the most commonly used (53.2%).

After PCI, oral anti-platelet drugs were used in this group (initially ticlopidine, then clopidogrel, prasugrel, or ticagrelor). Baseline demographic and clinical characteristics, angiographic findings, in-hospital data and drugs at discharge, depending on the use of GPIs, are presented in Table 1 and Table 2. The most significant differences between the groups were as follows: more advanced age of patients who were not treated with GPIs, less frequent diagnosis of STEMI, and much less often used IABP in this group. LM disease was diagnosed more often in patients who were treated with GPIs and the revascularization effect measured in the TIMI flow grade was worse in this group. Left ventricular ejection fraction was not different between the groups.

### 3.2. In-Hospital Adverse Events and 12-Month Mortality after Admission

In both groups, the same rates of in-hospital adverse events (stroke, subsequent MI, bleeding requiring blood transfusion) were observed. In patients treated with GPIs, cardiac arrest occurred less frequently before admission to the department, whereas this complication occurred more frequently in this group during hospitalization. In the whole study population, in-hospital death occurred in 42.1% of patients, regardless of the cause, and no statistically significant differences were found between the groups depending on the use of GPIs [OR: 0.97, 95% CI: 0.9–1.06; *p* = 0.53]. Twelve months after admission, a lower unadjusted mortality rate was reported in the group treated with GPIs (Table 2 and Figure 3).

### 3.3. Predictors of 12-Month Mortality

In the 12-month follow-up, CS in the course of STEMI, higher SBP on admission, hyperlipidemia, history of smoking, therapy with GPIs (*p* < 0.001) and higher LVEF were independent factors reducing the risk of death from any cause. The use of GPIs reduced the risk of 12-month overall mortality by approximately 17.3% in the group (Table 3). This benefit can also be confirmed by the analysis of the ROC curve (Figure 2). Independent risk factors increasing the risk of death from any cause during one year of the follow-up included higher age, previous stroke or MI, history of PAD or CKD, higher HR on admission, cardiac arrest during hospitalization, lower TIMI flow after PCI in IRA, and in-hospital inotropic and IABP support (Table 3).

### 3.4. Mortality and Predictors of 12-Month Mortality after Discharge

In the analyzed group, 5903 patients were discharged alive (Figure 1). Used Kaplan–Meier model suggest better 12 months survival of patients treated in-hospital with intravenous GPIs (Figure 4). In-hospital use of GPIs remained predictor reducing the 12 months mortality risk (Table 4). Oral antiplatelet drugs (ticlopidine, clopidogrel, prasugrel or ticagrelor) and the year of inclusion in the analysis had no impact on it.

## 4. Discussion

The above observational data on a large group of real-world patients may suggest that the addition of GPIs to the standard pharmacotherapy combined with PCI in patients with MI-induced CS on admission reduced the risk of death in the 12-month follow-up. In our observational study, in-hospital and 12-month mortality rates were 42.5% and 54.9%, respectively, in the group treated with GPIs. The use of these drugs was associated with a significant reduction in the risk of all-cause mortality in the 12-month follow-up since admission and after discharge compared to the group in which GPIs were not used.

Similar conclusions can be drawn from the analyses of small groups of patients treated with abciximab in the course of CS as reported by Antoniucci et al. (77 patients) and Chan et al. (observational study on 99 patients). In turn, the influence of eptifibatide on the course of CS induced by NSTEMI was assessed by Hasdai et al. [16,17,18]. In two of the largest observational studies, the addition of GPIs to the therapy was also associated with a lower risk of death in the short-term and during the 1-year follow-up [11,12]. All these studies and the meta-analysis of these data [19] may suggest the benefits of GPIs in the treatment of CS.

During enrollment for the analysis, recommendations for the use of GPIs changed over the years and were included in subsequent guidelines of cardiology societies. The main indication for their use is still the severity of coronary artery disease, including the location of the lesions and coronary artery blood flow assessed according to the TIMI scale. This was reflected in our observation—GPIs were more often used in high-risk patients with multivessel coronary artery disease, left main or proximal left anterior descending artery disease, or when the TIMI flow was worse in the IRA. The group of patients treated with GPIs was younger than those who did not receive GPIs and were at a lower risk of heart failure, but had lower blood pressure and higher HR on admission, had more advanced changes in coronary arteries and worse PCI outcome (according to the TIMI score). These are important parameters of an unfavorable prognosis in course of CS. Interestingly, the year of hospitalisation had no impact on 12-months prognosis after discharge (Figure 4).

Unfortunately, there are no exact data concerning the risk of bleeding complications or the use of GPIs from randomized trials or meta-analyses available in patients with CS [19]. Difficulties in data comparison are also associated with the change in definitions of bleeding complications in subsequent papers. In our study population, the use of GPIs had no effect on in-hospital hemorrhagic complications (2.6% vs. 2.9%). Similar results were observed by Kanic et al. [12], who found no differences in the incidence of minor or major bleeding. Conversely, De Felice et al. [11] found that hemorrhagic complications affected only 1% of patients treated with abciximab and did not differ statistically compared with the control group. In the analysis of Shugman et al. [20], which included 10% of CS, no differences were found in relation to the use of abciximab or tirofiban in the incidence of bleeding complications. That multivariable analysis did not show an increased risk of major bleeding complications related to the administration of abciximab or tirofiban, whereas the occurrence of CS symptoms and the female gender had an impact on the risk.

The beneficial effect of GPIs on the prognosis in the course of CS may result from several reasons. First, the administration of GPIs causes long-term passivation of atherosclerotic plaques in coronary arteries. Hasdai et al. assumed that this effect could persist up to 36 months [18]. Secondly, this group of drugs improves microcirculation, which is significantly impaired in CS by unfavorable hemodynamic conditions and additionally the overlapping microembolization caused by PCI. The worse bioavailability of oral antiplatelet agents in the course of CS can also be significant. We noted that simultaneous administration of oral and intravenous antiplatelet drugs (GPIs) reduced the risk of mortality. This situation does not occur in hemodynamically stable patients. In that case, the combination of oral and intravenous antiplatelet therapy is related to a higher risk of bleeding, as mentioned above.

### Limitations of the Study

The above analysis has some limitations. Since this retrospective analysis was an observational study, all limitations associated with a nonrandomized setting were also applicable. Definitions adopted in different studies made the comparison more difficult. Our data were collected over a 16-year period, in which treatment strategies, PCI techniques, and medication given during this procedure underwent some changes. Guidelines were not strictly adhered to in every case and therapy was individualized, if needed. Decisions regarding treatment methods, especially the use of intravenous GPIs, were left to the discretion of the attending physicians. We have no certain data on the compliance of recommended pharmacotherapy after discharge. Long-term data included only all-cause mortality, and information about death from cardiovascular cause was not available.

## 5. Conclusions

In our ‘real-world’ patients with MI and CS on admission, the addition of intravenous GPIs to the standard pharmacotherapy combined with PCI resulted in lower mortality in the 12-month follow-up without increasing in-hospital adverse events rates.

## Figures and Tables

**Figure 1 jcm-10-05059-f001:**
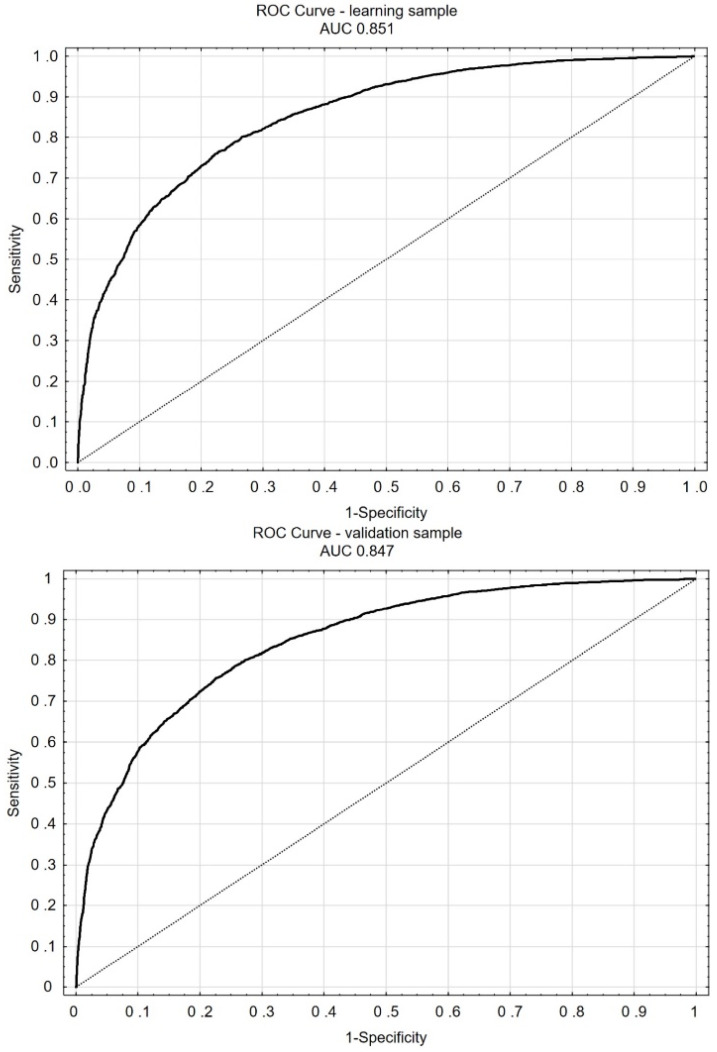
ROC curve for the multivariate model to assess the relationship between the use of glycoprotein IIb/IIIa receptor inhibitors in the cohort and 12-month mortality.

**Figure 2 jcm-10-05059-f002:**
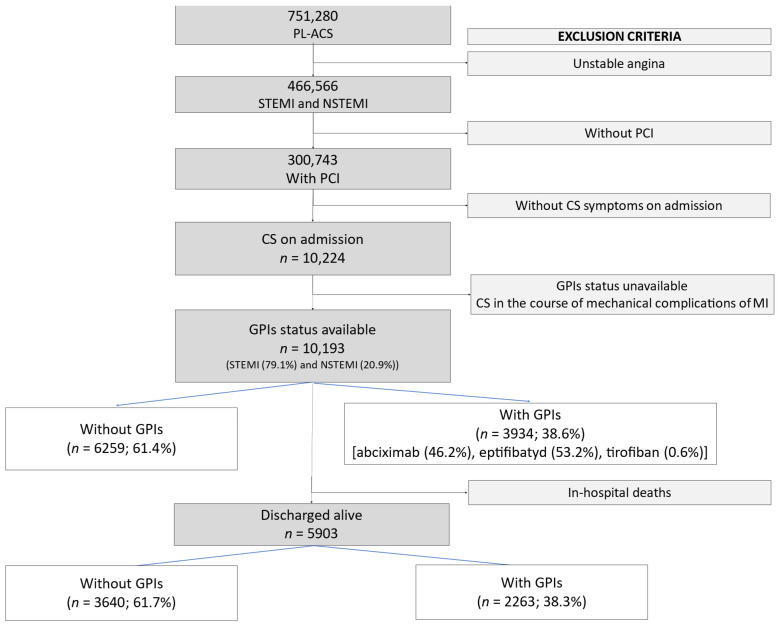
Study flow chart. CS—cardiogenic shock; GPIs—glycoprotein IIb/IIIa receptor inhibitors; MI—myocardial infarction; NSTEMI—non-ST-segment elevation myocardial infarction; PCI—percutaneous coronary intervention; PL-ACS—Polish Registry of Acute Coronary Syndromes; STEMI—ST-segment elevation myocardial infarction.

**Figure 3 jcm-10-05059-f003:**
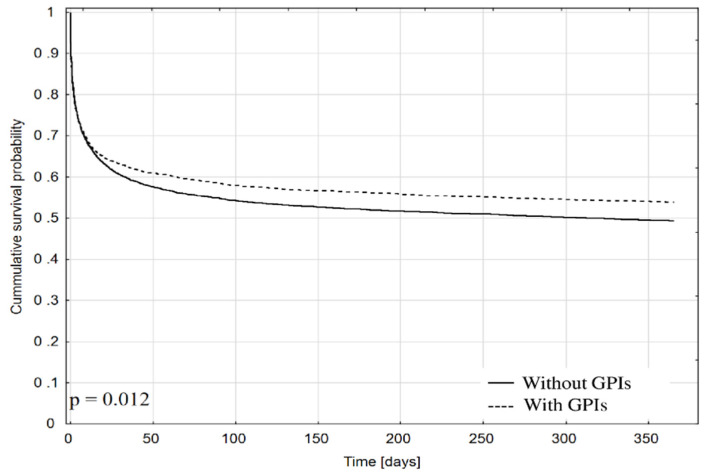
Kaplan–Meier curve showing cumulative survival probability since admission depending on the use of GPIs. GPIs—glycoprotein IIb/IIIa receptor inhibitors.

**Figure 4 jcm-10-05059-f004:**
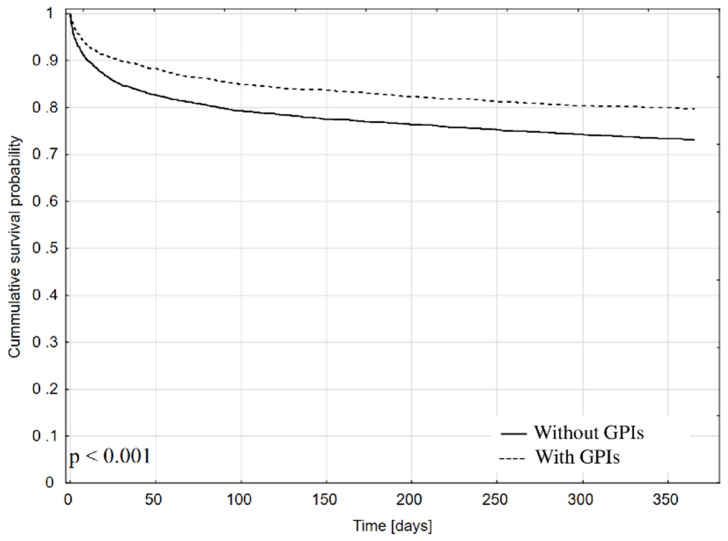
Kaplan–Meier curve showing cumulative survival probability in patients discharged alive depending on the in-hospital use of GPIs.

**Table 1 jcm-10-05059-t001:** Clinical characteristics, in-hospital data, and angiographic findings of patients depending on the use of GPIs.

		GPIs (−)	GPIs (+)	*p* Value
*N*		6259	3934	
Age, years		69.4 (11.8)	66.5 (11.7)	<0.001
Sex, males %		61.3	63.4	0.03
STEMI, %		74.7	86.1	<0.001
Arterial hypertension, %		56.8	54.7	0.047
Diabetes, %		29.5	27.4	0.02
History of smoking, %		49.7	54.5	<0.001
Hyperlipidemia, %		30.4	32.7	0.53
Previous MI, %		20.9	19.2	0.03
Previous PCI, %		10.5	11.4	0.17
Previous CABG, %		4.0	3.7	0.5
Chronic heart failure, %		12.7	9.5	<0.001
History of atrial fibrillation, %		12.4	10.7	0.01
Previous stroke, %		6.9	4.8	<0.001
History of PAD, %		7.5	6.8	0.23
History of CKD, %		13.1	9.9	<0.001
Cardiac arrest before admission, %		21.8	18.2	<0.001
In-hospital data				
SBP on admission, mmHg		90.0 (33.0)	82.0 (30.0)	<0.001
DBP on admission, mmHg		60.0 (20.0)	60.0 (21.0)	<0.001
MAP on admission, mmHg		70.0 (26.7)	66.7 (21.3)	<0.001
HR on admission, bpm		80.0 (32.0)	85.0 (40.0)	0.001
Inotrope use, %		37.9	45.2	<0.001
IABP use, %		11.2	20.1	<0.001
Unfractionated heparin use, %		96.9	97.8	0.01
Low-molecular-weight heparin use, %		11.9	10.4	0.048
Clopidogrel, %		83.8	84.5	0.32
Ticlopidine, %		7.6	7.3	0.56
Ticagrelor, %		3.8	4.9	<0.001
Prasugrel, %		1.7	2.1	0.18
LVEF, %		38 (20)	38 (21)	0.56
Angiographic findings				
LM disease, %		9.1	13.5	<0.001
MVD, %		35.5	37.5	0.046
LAD disease, %		44.4	48.6	<0.001
TIMI flow before PCI, %	0	61.3	74.0	<0.001
	1	14.0	11.4	
	2	10.8	6.8	
	3	13.9	7.8	
TIMI flow after PCI, %	0	9.7	7.0	<0.001
	1	4.7	6.6	
	2	8.9	14.3	
	3	76.7	72.1	

CKD—chronic kidney disease; DBP—diastolic blood pressure; GPIs—glycoprotein IIb/IIIa receptor inhibitors; HR—heart rate; IABP—intra-aortic balloon pump; LAD—left anterior descending artery; LM—left main; LVEF—left ventricular ejection fraction; MAP—mean arterial pressure; MI—myocardial infarction; MVD—multivessel disease; PAD—peripheral artery disease; PCI—percutaneous coronary intervention; SBP—systolic blood pressure; STEMI—ST-segment elevation MI; TIMI—thrombolysis in myocardial infarction.

**Table 2 jcm-10-05059-t002:** In-hospital adverse events, data at discharge and 12-month mortality of patients, depending on the use of GPIs.

	GPIs (−)	GPIs (+)	*p* Value
*N*	6259	3934	
MI during hospitalization, %	5.4	5.2	0.75
Stroke during hospitalization, %	0.9	0.9	0.88
Major bleeding during hospitalization (PL-ACS), %	2.6	2.9	0.44
Cardiac arrest during hospitalization, %	25.1	30.4	<0.001
In-hospital mortality, %	41.8	42.5	0.53
NYHA at discharge, %			
I	22.1	23.3	0.84
II	24.8	24.4	
III	13.4	11.5	
IV	39.7	40.8	
Drugs at discharge			
ASA, %	88.0	90.1	0.01
Second antiplatelet drug, %	76.3	80.6	<0.001
ACEI/ARB/ARNI, %	59.0	64.9	<0.001
Beta-blocker, %	66.9	70.7	0.002
Diuretic, %	35.6	37.3	0.18
Statin, %	76.5	81.2	<0.001
MRA, %	16.7	17.7	0.75
12-month mortality, %	57.9	54.9	0.002

ACE-I—angiotensin-converting enzyme inhibitor; ARB—angiotensin-receptor blocker; ARNI—angiotensin receptor neprilysin inhibitor; ASA—acetylsalicylic acid; GPIs—glycoprotein IIb/IIIa receptor inhibitors; MI—myocardial infarction; MRA—mineralocorticoid receptor antagonist; NYHA—New York Heart Association scale; PL-ACS—Polish Registry of Acute Coronary Syndromes.

**Table 3 jcm-10-05059-t003:** Factors affecting 12-month mortality in the multivariate analysis since admission.

Variables	OR (95% CI)
Age (for each 1-year increase)	1.038 (95% CI: 1.033–1.043); *p* < 0.001
History of hyperlipidemia	0.744 (95% CI: 0.667–0.831); *p* < 0.001
History of stroke	1.42 (95% CI: 1.1–1.833); *p* = 0.007
History of PAD	1.705 (95% CI: 1.339–2.171); *p* < 0.001
History of CKD	1.475 (95% CI: 1.22–1.783); *p* < 0.001
History of smoking	0.83 (95% CI: 0.74–0.92); *p* < 0.001
History of myocardial infarction	1.141 (95% CI: 1.004–1.297); *p* = 0.044
SBP on admission (for each 1 mmHg increase)	0.995 (95% CI: 0.993–0.996); *p* < 0.001
Heart rate on admission (for each 1 beat increase)	1.007 (95% CI: 1.005–1.008); *p* < 0.001
LVEF (for each 1% increase)	0.959 (95% CI: 0.956–0.961); *p* < 0.001
Inotropic drug use	1.59 (95% CI: 1.415–1.807); *p* < 0.001
IABP	1.748 (95% CI: 1.415–1.807); *p* < 0.001
GPIs use	0.827 (95% CI: 0.745–0.919); *p* < 0.001
TIMI 0 after PCI (vs. TIMI 3)	3.807 (95% CI: 3.035–4.776); *p* < 0.001
TIMI 1 after PCI (vs. TIMI 3)	3.687 (95% CI: 2.795–4.864); *p* < 0.001
TIMI 2 after PCI (vs. TIMI 3)	2.015 (95% CI: 1.701–2.388); *p* < 0.001
Cardiac arrest during hospitalization	3.667 (95% CI: 3.232–4.161); *p* < 0.001
STEMI (vs. NSTEMI)	0.771 (95% CI: 0.68–0.874); *p* < 0.001

CI—confidence interval; CKD—chronic kidney disease; GPIs—glycoprotein IIb/IIIa receptor inhibitors; HR—heart rate; IABP—intra-aortic balloon pump; LVEF—left ventricular ejection fraction; NSTEMI—non-ST-segment elevation MI; OR—odds ratio; PAD—peripheral artery disease; PCI—percutaneous coronary intervention; SBP—systolic blood pressure; STEMI—ST-segment elevation MI; TIMI—thrombolysis in myocardial infarction.

**Table 4 jcm-10-05059-t004:** Factors affecting 12-month mortality in the multivariate analysis in the group of patients discharged alive.

Variables	OR (95% CI)
Age (for each 1-year increase)	1.029 (95% CI: 1.022–1.035); *p* < 0.001
History of hyperlipidemia	0.727 (95% CI: 0.627–0.844); *p* < 0.001
History of COPD	1.617 (95% CI: 1.186–2.204); *p* = 0.002
History of PAD	1.478 (95% CI: 1.091–2.002); *p* = 0.012
Previous PCI	1.353 (95% CI: 1.096–1.671); *p* = 0.005
STEMI (vs. NSTEMI)	0.821 (95% CI: 0.701–0.961); *p* = 0.014
NYHA III or IV on admission	2.191 (95% CI: 1.847–2.600); *p* < 0.001
Heart rate on admission (for each 1 beat increase)	1.007 (95% CI: 1.005–1.009); *p* < 0.001
LVEF < 20%	3.702 (95% CI: 2.448–5.599); *p* < 0.001
LVEF 20–34%	2.685 (95% CI: 2.116–3.407); *p* < 0.001
LVEF 35–49%	1.548 (95% CI: 1.231–1.947); *p* < 0.001
No LVEF data	2.374 (95% CI: 1.875–3.006); *p* < 0.001
Stroke during hospitalization	2.253 (95% CI: 1.287–3.946); *p* = 0.004
In-hospital IABP use	1.436 (95% CI: 1.187–1.737); *p* < 0.001
In-hospital GPIs use	0.762 (95% CI: 0.662–0.878); *p* < 0.001
In-hospital beta-blocker use	0.908 (95% CI: 0.751–1.098); *p* = 0.32
In-hospital insulin use	1.510 (95% CI: 1.267–1.800); *p* < 0.001
In-hospital statin use	0.767 (95% CI: 0.640–0.918); *p* = 0.004
Beta-blocker at discharge	0.691 (95% CI: 0.586–0.815); *p* < 0.001
Years of hospitalization 2003–2006 (vs. 2016–2019)	0.708 (95% CI: 0.473–1.059); *p* = 0.093
Years of hospitalization 2007–2009 (vs. 2016–2019)	0.994 (95% CI: 0.719–1.375); *p* = 0.971
Years of hospitalization 2010–2012 (vs. 2016–2019)	0.708 (95% CI: 0.473–1.059); *p* = 0.093
Years of hospitalization 2013–2015 (vs. 2016–2019)	1.237 (95% CI: 0.935–1.636); *p* = 0.136

CI—confidence interval; COPD—chronic obstructive pulmonary disease; GPIs—glycoprotein IIb/IIIa receptor inhibitors; IABP—intra-aortic balloon pump; LVEF—left ventricular ejection fraction; NSTEMI—non-ST-segment elevation MI; NYHA—New York Heart Association; OR—odds ratio; PAD—peripheral artery disease; PCI—percutaneous coronary intervention; STEMI—ST-segment elevation MI.

## Data Availability

This study did not report of such data.

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
