# Peer review of "One-Year Outcome of Glycoprotein IIb/IIIa Inhibitor Therapy in Patients with Myocardial Infarction-Related Cardiogenic Shock"

_jcm, 2021, doi:10.3390/jcm10215059_

Round 1

Reviewer 1 Report

Refrain from using acronyms like MI and CS without spelling it out when first used.

I recommend the conclusion to be rephrased and toned down, as I am not sure this retrospective partially unadjusted analysis is powerful enough to result in such a conclusive language.

The conclusion this is brought in the first paragraph of the discussion assumes there is a definite casualty between the addition of GPIs to standard therapy and better outcomes. I suggest this to be rephrased and only describe the association witnessed in this analysis which is very sensitive to several biases and is far from being flawless. It is not only the retrospective character of the analysis which is an inherent limitation using this registry, but  also the decision how to present and discuss these findings that could be improved in my opinion.

I presume that during the last two decades there has been some fundamental changes in the recommended approach to the treatment of patients in CS. To name just a few - one could remember that the vastly used IABP in this context is far less recommended nowadays, while the use of novel MCSs has gained some popularity and during the last several years ECMO has also been used more frequently in this setting. This type of therapeutic intervention are usually reserved for the patients that are less stable hemodynamically, which might overlap with the group of patients treated with GPI in your study (as could be estimated from the baseline characteristics). I raise this point as an example of possible confounder, among others I did not mention, which needs to be considered in the interpretation of your results. It is possible that GPI treatment is merely a marker of patients with more severe presentation at admission, and therefore a surrogate marker for more aggressive therapeutic interventions and monitoring

Action Items that are essential to improve this analysis:

  1. Time-dependent analysis comparing the effect of period on results.
  2. Kaplan-Meier curves including landmark analysis at 1-week or 1-month and then 12 months
  3. provide the adjusted OR for in-hospital and 12-months mortality 
  4. Provide the type of anti-platlets used and analyze the impact of the use of these agents on the results
  5. Discuss and explain how the results that show identical 1-month mortality rate but lower 12-months rate could be explained. 
  6. No casualty should be implied as this is a very limited retrospective analysis prone to many biases and the very inconsistent results should be addressed with much more caution.  

Author Response

Dear, Reviewer #1

At the beginning we would like to thank you for your constructive comments on our paper. We appreciate the time and effort that you have dedicated to providing your valuable feedback on our manuscript. We have been able to incorporate changes to reflect most of suggestions. We have highlighted the newest changes within the manuscript.

Response to the reviewer' comments:

Comment 1: The suggestion regarding time-dependent analysis comparing the effect of period on results.

Response: In the analysis, consecutive patients between October 1, 2003, and August 1, 2019 with STEMI and NSTEMI related CS on admission who underwent PCI, as the final method of reperfusion treatment, were included. We divided the patients into five groups, sequentially according to inclusion years. Assignment to the consecutively obtained group, was used as variable in the multivariate analysis (Table 4). In conducted analysis, years of the hospitalization, didn’t affect on the 12-month mortality risk.

Comment 2: The suggestion on the conduct of Kaplan-Meier curves including landmark analysis at 1-week or 1-month and then 12 months.

Response:  An analysis of the cumulative survival probability depending on the use of GPIs has been attached – Figure 3 and Figure 4. It suggested a significant difference in favour of patients treated with GPIs.

Comment 3: The suggestion regarding provide the adjusted OR for in-hospital and 12-months mortality.

Response: Data were calculated and provided into the revised manuscript.

Comment 4: Provide the type of anti-platlets used and analyze the impact of the use of these agents on the results.

Response: We provided the type of anti-platelets drugs in the Table 1. Type of  anti-platelets drug was used as variable in the multivariate analysis (Table 4). In conducted analysis, we didn't found an association between the used antiplatelet drug and the 12-month mortality risk.

Comment 5: Discuss and explain how the results that show identical 1-month mortality rate but lower 12-months rate could be explained. 

Response: In this study, we present the results of retrospective analysis of a data comming from the prospective national registry of acute coronary syndromes covering patients treated in Poland (PL-ACS). All included patients, 10,193, presented CS symptoms on admission induced by myocardial infarction. All patients underwent percutaneous revascularization. During the procedures in some patients GPIs were provided. Decision related to the use of  intravenous GPIs were left to the discretion of the attending physicians. Patients who received GPIs were predominantly those with severe symptoms of CS, lower blood pressure on admission, more advanced coronary artery disease and worse preprocedural coronary flow in the IRA assessed by the TIMI score. These are important parameters of an unfavorable prognosis in course of CS. Frequently use of GPIs in these clinical indications may be associated with an observed benefits at 12-month follow-up (Figure 3 and Figure 4). This fact may support the results of attached multivariate analysis (Table 3 and Table 4). We should also mention, the publications of Kanic et al. 2017 and the meta-analysis of Saleiro et al. 2020 which also highlighted the possible improvement in prognosis of patients treated with additional GPIs.

Several potential mechanisms can be considered to explain potential benefits of GPIs. First, the administration of GPIs cause long-term passivation of atherosclerotic plaques in the coronary arteries. Secondly, this group of drugs improves microcirculation, which is significantly impaired in CS by unfavorable hemodynamic conditions and additionally the overlapping microembolization caused by PCI. GPIs blockade may relieve the microvascular obstruction in the shock patients, resulting in improved prognosis. The worse bioavailability of oral antiplatelet agents in the course of CS can also be significant. This situations does not occur in the haemodynamically stable patients.

Comment 6: No casualty should be implied as this is a very limited retrospective analysis prone to many biases and the very inconsistent results should be addressed with much more caution.  

Response: Thank you for this valuable comment. The conclusion have been revised and more carefully worded in the submitted manuscript.

We look forward to hearing from you regarding our submission and respond to any further questions and comments you may have.

Sincerely,

Krzysztof Myrda

Reviewer 2 Report

I read the article with a great interest. The authors conclude that addition of GPIs combined with PCI improves the 12-month prognosis of patients with AMI and cardiogenic shock.

I have just two remarks.

  1. It would be interesting to comment how GPIs were given to the patients. If i.c. or i.v., just boluses or prolong infusions? Do You have any data about it?

2. The inhospital mortality was the same 41.8% vs 42.5% (GPI- vs GPI+), but 12M mortality different 57.9% vs 54.9% (p=0.002). I would appreciate the comment if it cannot be explained by different drugs at discharge. Patients in GPIs+ group had clearly better medication at discharge and probably during follow-up than in GPI-. And we know that this medication is very effective for prognosis of the patients with heart failure.

Author Response

Dear, Reviewer #2

At the beginning we would like to thank you for your constructive comments on our paper. We appreciate the time and effort that you have dedicated to providing your valuable feedback on our manuscript. We have been able to incorporate changes to reflect most of suggestions. We have highlighted the newest changes within the manuscript.

Response to the reviewer' comments:

Comment 1: It would be interesting to comment how GPIs were given to the patients. If i.c. or i.v., just boluses or prolong infusions? Do You have any data about it?

Response: The standard use of GPIs involves dosing based on patient’s weight; initial bolus i.v. and then continuous i.v. infusion. However, there is no accurate data on this details in the collected database. 

Comment 2: The inhospital mortality was the same 41.8% vs 42.5% (GPI- vs GPI+), but 12M mortality different 57.9% vs 54.9% (p=0.002). I would appreciate the comment if it cannot be explained by different drugs at discharge. Patients in GPIs+ group had clearly better medication at discharge and probably during follow-up than in GPI-. And we know that this medication is very effective for prognosis of the patients with heart failure.

Response: In this study, we present the results of retrospective analysis of a data comming from the prospective national registry of acute coronary syndromes covering patients treated in Poland (PL-ACS). All included patients, 10,193, presented CS symptoms on admission induced by myocardial infarction. All patients underwent percutaneous revascularization. During the procedures in some patients GPIs were provided. Decision related to the use of  intravenous GPIs were left to the discretion of the attending physicians. Patients who received GPIs were predominantly those with severe symptoms of CS, lower blood pressure on admission, more advanced coronary artery disease and worse preprocedural coronary flow in the IRA assessed by the TIMI score. These are important parameters of an unfavorable prognosis in course of CS. Frequently use of GPIs in these clinical indications may be associated with an observed benefits at 12-month follow-up (Figure 3 and Figure 4). This fact may support the results of attached multivariate analysis (Table 3 and Table 4). We should also mention, the publications of Kanic et al. 2017 and the meta-analysis of Saleiro et al. 2020 which also suggested the possible improvement in prognosis of patients treated with additional GPIs.

Several potential mechanisms can be considered to explain potential benefits of GPIs. First, the administration of GPIs cause long-term passivation of atherosclerotic plaques in the coronary arteries. Secondly, this group of drugs improves microcirculation, which is significantly impaired in CS by unfavorable hemodynamic conditions and additionally the overlapping microembolization caused by PCI. GPIs blockade may relieve the microvascular obstruction in the shock patients, resulting in improved prognosis. The worse bioavailability of oral antiplatelet agents in the course of CS can also be significant. This situations does not occur in the haemodynamically stable patients.

The improvement in long-term prognosis can be related to the medication at discharge. It also suggest our analysis (Table 4). Unfortunately, we have no certain data on the compliance of recommended pharmacotherapy after discharge.

We look forward to hearing from you regarding our submission and respond to any further questions and comments you may have.

Sincerely,

Krzysztof Myrda
